# Radical Pairing Interactions and Donor–Acceptor Interactions in Cyclobis(paraquat-p-phenylene) Inclusion Complexes

**DOI:** 10.3390/molecules28052057

**Published:** 2023-02-22

**Authors:** Wei Wang, Wei Wu, Peifeng Su

**Affiliations:** Fujian Provincial Key Laboratory of Theoretical and Computational Chemistry, The State Key Laboratory of Physical Chemistry of Solid Surfaces, and College of Chemistry and Chemical Engineering, Xiamen University, Xiamen 361005, China

**Keywords:** radical pairing interactions, donor–acceptor interactions, mechanically interlocked molecules, energy decomposition analysis

## Abstract

Understanding molecular interactions in mechanically interlocked molecules (MIMs) is challenging because they can be either donor–acceptor interactions or radical pairing interactions, depending on the charge states and multiplicities in the different components of the MIMs. In this work, for the first time, the interactions between cyclobis(paraquat-p-phenylene) (abbreviated as CBPQT^n+^ (*n* = 0–4)) and a series of recognition units (RUs) were investigated using the energy decomposition analysis approach (EDA). These RUs include bipyridinium radical cation (BIPY^•+^), naphthalene-1,8:4,5-bis(dicarboximide) radical anion (NDI^•−^), their oxidized states (BIPY^2+^ and NDI), neutral electron-rich tetrathiafulvalene (TTF) and neutral bis-dithiazolyl radical (BTA^•^). The results of generalized Kohn–Sham energy decomposition analysis (GKS-EDA) reveal that for the CBPQT^n+^···RU interactions, correlation/dispersion terms always have large contributions, while electrostatic and desolvation terms are sensitive to the variation in charge states in CBPQT^n+^ and RU. For all the CBPQT^n+^···RU interactions, desolvation terms always tend to overcome the repulsive electrostatic interactions between the CBPQT cation and RU cation. Electrostatic interaction is important when RU has the negative charge. Moreover, the different physical origins of donor–acceptor interactions and radical pairing interactions are compared and discussed. Compared to donor–acceptor interactions, in radical pairing interactions, the polarization term is always small, while the correlation/dispersion term is important. With regard to donor–acceptor interactions, in some cases, polarization terms could be quite large due to the electron transfer between the CBPQT ring and RU, which responds to the large geometrical relaxation of the whole systems.

## 1. Introduction

Mechanically interlocked molecules (MIMs) have been widely used to develop various molecular machines, including molecular switches [1,2,3,4,5], drug delivery systems [6,7], artificial molecular muscles [8,9,10], etc. MIMs are composed of multiple mechanically interlocked components connected by a mechanical bond, which can be described as an entanglement in space that prevents two parts of a molecule from separating. By modulating the non-covalent interactions between different components of MIMs, mechanical movement could occur. Pioneered by Stoddart et al., cyclobis(paraquat-p-phenylene) (CBPQT^4+^) has been widely used in MIMs because of its unique redox properties [11,12]. It is known that different charge states of CBPQT lead to the different kinds of interactions, so understanding the non-covalent interactions between CBPQT^n+^ (*n* = 0–4) and recognition units (RU) is important for the design of MIMs. In a fully oxidized state, electron-deficient tetracationic cyclophane (CBPQT^4+^) prefers to reside on a strong π-electron donor. Caramori and Muñoz-Castro [13] summarized the nature of host–guest interactions between CBPQT^4+^ derivatives and a variety of closed-shell guests, showing that these CBPQT^4+^···RU interactions are dispersion interactions. They also pointed out that the variation in such interactions with the different charge states of CBPQT have not been investigated in detail.

Most experimental and theoretical studies focus on the CBPQT^4+^···RU systems. Recently, several works have been devoted to exploring the role of radical pairing interactions as the driving forces in molecular switches [14,15,16,17,18,19]. For example, because of two unpaired electrons, CBPQT^2(•+)^ is no longer π-electron-deficient enough to undergo donor–acceptor interactions. Thus, in CBPQT^2(•+)^ complexes, radical pairing interactions could happen when the RU is radical. Stoddart et al. [14] found that there was extra covalent bonding between CBPQT^2(•+)^ and bipyridinium radical cation (BIPY^•+^). Stoddart et al. [12] stated that in the future directions for material design, the role of radical pairing interaction should be highly anticipated. Recently, Li et al. reported that Coulombic attraction is more important in the formation of the host–guest complex consisting of CBPQT^2(•+)^ and naphthalene-1,8:4,5-bis(dicarboximide) radical anion (NDI^•−^) [20]. Given the several special cases mentioned above, the nature of radical pairing interactions with various CBPQT and RU deserve further exploration.

In this work, our motivation is to investigate the nature of various interactions between CBPQT^n+^ (*n* = 0–4) and different types of RUs. The selection of RU would take the diversity of multiplicity and charge into account. The RUs in this work include bipyridinium radical cation (BIPY^•+^), bipyridinium dication (BIPY^2+^), naphthalene-1,8:4,5-bis(dicarboximide) NDI and its radical anion NDI^•−^, bis-dithiazolyl radical (BTA^•^) and tetrathiafulvalene (TTF). Thus, these RUs can be grouped into two categories, closed-shell RUs (BIPY^2+^, NDI and TTF) and open-shell RUs (BIPY^•+^, NDI^•−^ and BTA^•^). 

Energy decomposition analysis (EDA) methods are widely used to provide quantitative interpretation for non-covalent interactions. However, current EDA theoretical studies for interactions in MIMs focus on CBPQT^4+^ and closed-shell RUs, while those for radical–radical interactions between CBPQT^n+^ and open-shell RUs are quite limited. This is because the theoretical analysis for radical–radical interactions is difficult for most of the current EDA methods. Recently, based on broken symmetry (BS) unrestricted density functional theory, an extension of GKS-EDA called GKS-EDA(BS) has been proposed for intermolecular interactions in an open-shell singlet. In GKS-EDA(BS), spin projection approximation is implemented to consider the spin contamination. When the value of <*S*^2^> goes from 1.0 to 0.0, GKS-EDA(BS) can smoothly retreat to GKS-EDA. This means that the GKS-EDA method is able to explore the nature of CBPQT^n+^···RU interactions with the variations of charge states and multiplicities to provide panoramic insight for the interactions in MIMs. 

## 2. Results and Discussion

### 2.1. Geometries of CBPQT^n+^ (n = 0–4) and RU···CBPQT^n+^ Complexes

Figure 1, Figure 2, Figure 3, Figure 4, Figure 5, Figure 6 and Figure 7 demonstrate the optimized geometries, the natural population analysis (NPA) charge distributions and spin density distributions of CBPQT^n+^ monomers and RU···CBPQT^n+^ complexes, respectively. In general, the optimized geometries accord with those in the literature quite well. In detail, from CBPQT^4+^ to CBPQT^0^, the distances between N and N’ atoms increase from 6.453 Å to 6.770 Å, while the dihedral angle between two rings of each bipyridinium (BIPY) unit varies with the charge on this unit, which is consistent with experimental results [16]. Moreover, the electronic ground states of CBPQT^4+^ and CBPQT^0^ are the closed-shell singlet; those of CBPQT^•3+^ and CBPQT^•+^ belong to the doublet state. In the case of diradical CBPQT^2(•+)^, the unrestricted ωB97X-D/6-31G(d) calculations indicate that the open-shell singlet and triplet state are degenerate (only 0.01 kcal/mol difference). The energy degeneracy of the open-shell singlet and triplet state is because of the long distance between two BIPY^•+^ units (larger than 6 Å) with one unpaired electron, consistent with the spin density distributions. 

For BIPY^2+^···CBPQT^n+^, the torsional angle of the BIPY^2+^ unit decreases significantly upon complexation with rings, except CBPQT^4+^, which is consistent with the results from the literature [21,22]. In TTF···CBPQT^4+^, the interplanar distances from TTF to the two BIPY^2+^ units on CBPQT^4+^ are close (3.620 Å and 3.609 Å, respectively), corresponding to the crystal data [23]. For BIPY^•+^···CBPQT^n+^, BIPY^•+^ remains near-planar upon complexation with rings in different oxidation states, which agrees with the results of Stoddart et al. [21,24].

According to the NPA results, in most of complexes, the charges of the host and the guest molecules are close to the corresponding isolated ones. However, the NPA charges in BIPY^2+^···CBPQT^•+/0^ and NDI···CBPQT^•+/0^ suggest partial electron transfer between host and guest. 

From the spin density distributions, in the closed-shell RU complexes, the unpaired spin is mainly located at the CBPQT ring, while in the open-shell RU complexes, unpaired spins are distributed in both the CBPQT ring and RU anti-parallelly, which indicates that these host–guest interactions belong to radical–radical interactions with the broken symmetry character. 

### 2.2. Interactions between Closed-Shell Recognition Units and CBPQT^n+^ in Solution

With the variation in *n* values, the interactions between closed-shell RU (CRU) and CBPQT^n+^ can be closed-shell···closed-shell or closed-shell···radical interactions. 

The AIM and IGM analysis plots of CRU···CBPQT^n+^ complexes are shown in Figure 8, Figure 9 and Appendix A, respectively. The BCP parameters collected in Appendix A illustrate that there are various interactions between the functional groups of the hosts and guests. The |*V*|/*G* values range from 0.60 to 0.84, showing that the CBPQT^n+^ and CRUs are non-covalently connected. In detail, for BIPY^2+^···CBPQT^n+^, there are π–π interactions and C-H···π interactions between BIPY^2+^ and the ring. For NDI···CBPQT^n+^, aside from π–π interactions, there are lone pair···π interactions between the O atom in NDI and p-xylylene moieties on the ring, and C-H···π interactions between C-H and p-xylylene moieties. For TTF···CBPQT^n+^, aside from π–π interactions, there are lone pair···π interactions between TTF and CBPQT^n+^. In general, the IGM analysis plots provide similar conclusions to the AIM results. Notably, the IGM plots suggest that the interactions in BIPY^2+^···CBPQT^0^ and BIPY^2+^···CBPQT^•+^ could be stronger than those in the other BIPY^2+^···CBPQT^n+^. 

Appendix A show the *D*_0_, ∆*G*^TOT^ and ∆*G*^ZPE+GEO^ values of the CRU···CBPQT^n+^ complexes. It is noted that a large volume of CRU or electron transfer would lead to the large ∆*G*^ZPE+GEO^ values. Despite BIPY^2+^···CBPQT^4+^, the dominant role of ∆*G*^TOT^ in *D*_0_ is confirmed. Next, the GKS-EDA results of ∆*G*^TOT^ are discussed, which are summarized in Table 1, Table 2 and Table 3. For BIPY^2+^···CBPQT^n+^, with the decrease in *n* value, the total interactions become large, which is mainly attributed to the reduction in the repulsive electrostatic term. It can be found that despite correlation/dispersion terms, the other EDA terms are very sensitive to the variation in charge states in the CBPQT ring. For *n* ≥ 2, the total interaction energies are dominated by desolvation term, showing the importance of solvent effects to overcome the large repulsive electrostatic interactions between two cations. Among these EDA terms, the values of the correlation/dispersion term are larger than polarization and considered to be the secondary contribution. For BIPY^2+^···CBPQT^•+/0^, the polarization makes an important contribution in agreement with the NPA analysis, which shows the large electron transfer between BIPY^2+^ and CBPQT^•+/0^. Furthermore, there is an excellent linear relationship between the variation in charge (∆*q*) on BIPY^2+^ and polarization interactions of the BIPY^2+^···CBPQT^n+^ complexes, indicating the correlation between polarization and electron transfer, as shown in Figure 10. The large electron transfer between BIPY^2+^ and CBPQT^•+/0^ indicates the covalent bonding character, in agreement with the large geometry relaxation compared to the other BIPY^2+^···CBPQT^n+^ complexes shown in Appendix A. 

For the NDI···CBPQT^n+^ interactions, it can be found that the correlation/dispersion terms are largest among the EDA terms, governing the interactions, while the electrostatic terms play the secondary role. The desolvation term shows that the influence of the solvent effect can be negligible. The contribution of the polarization term is smaller than the electrostatic and correlation/dispersion terms. Similar to BIPY^2+^···CBPQT^•+/0^, the polarization terms in NDI···CBPQT^•+/0^ are relatively large, which suggests electron transfer between NDI and CBPQT^•+/0^. 

Like NDI···CBPQT^n+^, the TTF···CBPQT^n+^ interactions are also dominated by correlation/dispersion terms. The contribution of the polarization term is small, denoting the non-covalent character of the interactions. The electrostatic interaction facilitates the total interactions, while the unfavorable desolvation term is small. It is noted that with the decrease in the positive charge on the CBPQT ring, the total interaction energy decreases, mainly because of the reduced electrostatic term. 

### 2.3. Interactions between Open-Shell Recognition Units and CBPQT^n+^ in Solution

The interactions between open-shell RU (ORU) and CBPQT^n+^ can be radical···closed-shell or radical···radical interactions, given the variation in charges and multiplicities. According to the results in the literature [14,17,20,25], the complex of CBPQT^n+^ and ORU with anti-parallel low spins is more stable than that with high spins. Our test calculations also confirm the stability of the ORU···CBPQT^n+^ complex in low-spin states (2.5 kcal/mol lower than the high-spin state of BIPY^•+^···CBPQT^•+^). Thus, in the radical pairing interactions discussed below, only low-spin states are considered. 

The AIM and IGM analysis results of ORU···CBPQT^n+^ are collected in Appendix A, while the BCP parameters from AIM results are listed in Appendix A. The values of *ρ*, ∇^2^*ρ* and |*V*|/*G* are similar to the CRU···CBPQT^n+^ interactions, showing that these interactions also belong to non-covalent interactions. In BIPY^•+^···CBPQT^n+^, there are π···π interactions and C-H···π interactions between BIPY^•+^ and the ring. For NDI^•−^···CBPQT^n+^ and BTA^•^···CBPQT^n+^, it is clear that there are π–π interactions, lone pair···π interactions and C-H···π interactions between NDI^•−^/BTA^•^ and the ring. 

The *D*_0_, ∆*G*^TOT^ and ∆*G*^ZPE+GEO^ values of the ORU···CBPQT^n+^ complexes are shown in Appendix A. Compared to the CRU complexes, in the ORU complexes, ∆*G*^TOT^ is always larger than ∆*G*^ZPE+GEO^, contributing the attractive interactions. Table 4, Table 5 and Table 6 collect the GKS-EDA results for the ORU···CBPQT^n+^ interactions. For BIPY^•+^···CBPQT^n+^, the total interaction energies range from −19.17 to −36.42 kcal/mol. The interaction energy of BIPY^•+^···CBPQT^2(•+)^, −33.8 kcal/mol, is close to the result of −29.6 kcal/mol in the literature [14]. In general, the physical origin of the total interactions in BIPY^•+^···CBPQT^n+^ is analogous to those of BIPY^2+^···CBPQT^n+^. The electrostatic and desolvation terms are sensitive to the variation in *n* value. From CBPQT^4+^ to CBPQT^0^, the host–guest interactions increase, which can be contributed to by the variation in the electrostatic term. Compared to BIPY^2+^···CBPQT^n+^, in BIPY^•+^···CBPQT^n+^, the role of correlation/dispersion becomes more important, while the contribution of the polarization term becomes smaller. Furthermore, it can be found that, in contrast to BIPY^2+^···CBPQT^•+/0^, the polarization term in BIPY^•+^···CBPQT^•+/0^ is still small. This means that the orbital relaxation, which contains the effects of charge transfer and induction, is always unimportant in BIPY^•+^···CBPQT^n+^.

To clarify the large differences in polarization terms in the BIPY^•+^ and BIPY^2+^ complexes, the orbital diagrams of BIPY^•+^···CBPQT^n+^ and BIPY^2+^···CBPQT^n+^ (*n* = 0 and 3) are displayed in Figure 11. It is shown that for the two BIPY^2+^ complexes, the orbital interactions are different. In BIPY^2+^···CBPQT^0^, there is strong orbital mixing between the LUMO of the BIPY^2+^ and the HOMO of the CBPQT^0^, while in BIPY^2+^···CBPQT^•3+^, the orbital mixing is weak. Moreover, in the two BIPY^•+^ complexes, the overlaps of the SOMO of the BIPY^•+^ and the HOMO of the CBPQT^n+^ are always weak. This is in agreement with the small polarization terms in Table 4. Overall, the variation in BIPY^•+^···CBPQT^n+^ interaction is weaker than that of BIPY^2+^, indicating that BIPY^2+^ is more sensitive to the charge state of the CBPQT ring than BIPY^•+,^ while the polarization interactions between BIPY^•+^ and the ring are negligible, so this may be the reason that BIPY^•+^ is widely used in molecular machines.

For NDI^•−^···CBPQT^n+^, the electrostatic term is the largest contribution, except for NDI^•−^···CBPQT^0^. In the NDI^•−^···CBPQT^0^ interaction, the total interaction energy is dominated by the correlation/dispersion term. From CBPQT^0^ to CBPQT^4+^, the electrostatic and desolvation terms are sensitive to the variation in the charged state, while the correlation/dispersion term is insensitive to this variation, playing another important stabilizing role in the total interactions. The contribution of the polarization term is small, indicating the weak orbital variation between NDI^•−^ and CBPQT^n+^. These results show that with a large *n* value, the binding for such heteroradical systems is greatly enhanced by electrostatic attraction, in agreement with the conclusion of Li et al. [20], while the correlation/dispersion terms play the comparable role even with the small *n* value. However, when *n* = 0, the correlation/dispersion term is larger than the electrostatic term, showing the dominant contribution. 

For BTA^•^···CBPQT^n+^, the GKS-EDA results in Table 6 reveal that, in contrast to the complexes discussed above, not only the total interaction energies but also the individual terms are insensitive to the variation in charged states in the ring. The dispersion and correlation terms are dominant in the interactions, while the electrostatic terms are also beneficial to the interactions. The values of the polarization and desolvation terms are always small. In general, the BTA^•^···CBPQT^n+^ interaction is not sensitive to the change in charge states in CBPQT. 

### 2.4. Interactions of Inclusion Complexes in Solution with Counterions 

In the discussions above, the solvent environment is considered by the implicit solvation model. It is largely different from that in a real system because in experiments, counterions such as PF_6_^−^ are widely employed to balance the charge of the CBPQT complexes. In this section, the effects of counterions on the interactions between the host and guest in solution are explored. The PF_6_^−^ anions were added to counterbalance the excess positive charge of CBPQT^2(•+)^, and their relative position was extracted from the crystal structures. The fully optimized structures of five aggregates containing different recognition units are shown in Figure 12. 

In order to compare with the two-body interactions discussed above, the ring and counterions are considered as the CBPQT^2(•+)^(PF_6_^−^)_n_^m^ complex (*n* is the number of PF_6_^−^ anions while m is the opposite charges to the guest molecule). The GKS-EDA results of the RU···CBPQT^2(•+)^(PF_6_^−^)_n_^m^ interactions are listed in Table 7. 

For the neutral RUs, the introduction of counterions does not change the origin of the interactions. According to the GKS-EDA results, compared to the corresponding interaction without counterions, the total interactions of NDI···CBPQT^2(•+)^(PF_6_^−^)_2_ and BTA^•^···CBPQT^2(•+)^(PF_6_^−^)_2_ are stronger because of the increase in Δ*G*^disp/corr^ and the decrease in Δ*G*^exrep^. These interactions are still governed by Δ*G*^disp/corr^. 

For the cation RUs BIPY^2+^ and BIPY^•+^, after adding counterions, the electrostatic terms become negative (attractive), while the dispersion and correlation terms are also enhanced, which leads to the increase in total interaction energy. 

For the radical anion NDI^•−^, the addition of counterions weakens the electrostatic attraction and slightly increases the exchange repulsion. Although the polarization, dispersion and correlation terms are enhanced, the desolvation term becomes smaller. Overall, the total interaction energy is almost unchanged compared to the interaction between NDI^•−^ and CBPQT^2(•+)^.

In general, for CBPQT^2(•+)^, the addition of counterions mainly affects the electrostatic, dispersion and correlation terms between the host and the guest, while it clearly does not change the polarization term. 

## 3. Methodology and Computational Details

Generalized Kohn–Sham energy decomposition analysis (GKS-EDA) [26] has been widely used to investigate intra- and intermolecular interactions in both gas phase and polarizable medium [27,28,29,30,31,32]. In GKS-EDA, the total interaction energy in gas phase can be decomposed into the following terms:∆*E*^TOT^ = ∆*E*^ele^ + Δ*E*^exrep^ + Δ*E*^pol^ + Δ*E*^corr^ + Δ*E*^disp^
(1)
where Δ*E*^ele^, Δ*E*^exrep^, Δ*E*^pol^, Δ*E*^corr^ and Δ*E*^disp^ are electrostatic, exchange-repulsion, polarization, correlation and dispersion terms, respectively. Δ*E*^ele^ is Coulomb interaction between the electrons and nucleus from different monomers. Δ*E*^exrep^ denotes the Pauli repulsion between monomers. Δ*E*^pol^ represents the contribution of the orbital relaxation in the SCF procedure. Δ*E*^corr^ denotes the contribution of correlation energy from *E*_XC_ functionals and KS orbitals, defined as the difference in the exchange and correlation functionals, and the difference in exact exchange energies between the summation of monomers and supermolecule. Δ*E*^disp^ is optional, and is only available in dispersion correction DFT because the recently developed DFT functionals, especially for hybrid meta-GGA functionals such as M06-2X or the range-separated functional ωB97X-D, are capable of describing vdW interactions. Thus, the correlation term in these functionals tends to mimic the contribution of dispersion. In practice, Δ*E*^corr^ and Δ*E*^disp^ are often combined as Δ*E*^corr/disp^ term, called the correlation/dispersion term, to consider the long-range and short-range dispersion (or correlation) contribution when a dispersion correction functional such as ωB97X-D is applied. 

For intermolecular interactions in a solvent environment, GKS-EDA uses an implicit solvation model to consider the influence of solvent effects. The solvent environment is treated as a dielectric medium and polarized by the charge distribution of the solute molecule. The solute molecule is inserted into a cavity (or cavities) in the dielectric medium. The interaction between the solute charges and the polarized electric field of the solvent is represented as a reaction field operator and then put into a self-consistent reaction field (SCRF) procedure. Thus, instead of that in the gas phase, the GKS-EDA calculation in the solvent environment is performed with the wave function optimized by the SCRF procedure, denoted by Φ^SOL^. 

Considering a supermolecule S formed by several monomers A immersed in a polarizable medium, the total interaction free energy ∆*G*^TOT^ is expressed as:(2)$$\begin{array}{ll} \Delta G^{\mathrm{TOT}} & =G_{S}-\sum_{A}G_{A}\\ & =\Delta G^{\mathrm{ele}}+\Delta G^{\mathrm{exrep}}+\Delta G^{\mathrm{pol}}+\Delta G^{\mathrm{corr}}+\Delta G^{\mathrm{disp}}+\Delta G^{\mathrm{desol}} \end{array}$$

Here, *G*_S_ and *G*_A_ are the solvation free energies of supermolecule S and monomer A, respectively, which are computed by the implicit solvation model. The expressions of Δ*G*^ele^, Δ*G*^ex^, Δ*G*^rep^, Δ*G*^pol^ and Δ*G*^corr^ are the same as Δ*E*^ele^, Δ*E*^ex^, Δ*E*^rep^, Δ*E*^pol^ and Δ*E*^corr^, respectively. The notation ‘*G*’ is applied instead of ‘*E*’ because these interaction free energy terms are determined by Φ^SOL^. Δ*G*^desol^ reflects the influence of solvent environments computed by an implicit solvation model such as the CPCM method [33,34]. It accounts for the free energy penalty by the environment due to monomers’ interaction.

According to GKS-EDA(BS), the total interaction energy and the individual EDA terms can be calculated from the corresponding values in the broken symmetry (BS) singlet state and high-spin (HS) state:(3)$$\Delta G_{\mathrm{GS}}^{\mathrm{int}}=\Delta G_{\mathrm{GS}}^{\mathrm{ele}}+\Delta G_{\mathrm{GS}}^{\mathrm{exrep}}+\Delta G_{\mathrm{GS}}^{\mathrm{pol}}+\Delta G_{\mathrm{GS}}^{\mathrm{corr}/\mathrm{disp}}+\Delta G_{\mathrm{GS}}^{\mathrm{desol}}$$
where each energy component is obtained with the spin projection approximation proposed by Yamaguchi et al. [35,36]:(4)$$\Delta G_{\mathrm{GS}}^{X}=\left(1+c\right)\Delta G_{\mathrm{BS}}^{X}-c\Delta G_{\mathrm{HS}}^{X}$$
where *c* is defined as
(5)$$c=\frac{{\langle {\hat{S}}^{2}\rangle }_{\mathrm{BS}}}{{\langle {\hat{S}}^{2}\rangle }_{\mathrm{HS}}-{\langle {\hat{S}}^{2}\rangle }_{\mathrm{BS}}}$$

Here, ${\langle {\hat{S}}^{2}\rangle }_{\mathrm{BS}}$ and ${\langle {\hat{S}}^{2}\rangle }_{\mathrm{HS}}$ are the expectation values of BS and HS, respectively. When *c* = 0, the GKS-EDA(BS) scheme will retreat to the normal GKS-EDA.

Finally, the intermolecular binding energy (*D*_0_) can be defined as the absolute values of the combination of total interaction energy and Δ*G*^ZPE+GEO^. Δ*G*^ZPE+GEO^ is the geometrical relaxation and zero-point energy correction.
*D*_0_ = |Δ*G*^TOT^ + Δ*G*^ZPE+GEO^|(6)

In this work, all geometry optimizations and natural population analysis were performed using the Gaussian 09 program [37]. The range-separated functional ωB97X-D [38] with 6-31G(d) basis set was used for optimizations. The calculations of open-shell systems were carried out using broken symmetry unrestricted density functional theory. Atoms in molecules (AIM) [39] and independent gradient model (IGM) [40] were performed using the Multiwfn program [41] and visualized using VMD software [42].

GKS-EDA/GKS-EDA(BS) calculations were performed at the ωB97X-D/6-31+G(d) level using the XEDA program [43], interfaced with the GAMESS program [44]. In EDA calculations, the basis set superposition error (BSSE) was considered [45]. Solvent effect is considered through CPCM [33,34], in which the dielectric constant *ε* was set as 37.5 to model the acetonitrile solution, with reference to experimental and theoretical studies [16,17,46,47], and the cavity radii were determined using the universal force field (UFF) radii model scaled by a factor of 1.1 [48]. 

## 4. Conclusions

Thanks to the rapid development of MIMs, a comprehensive understanding of the radical pairing interaction in molecular switches is highly anticipated. In this article, the RU···CBPQT^n+^ (*n* = 0–4, RU = BIPY^2+^, NDI, TTF, BIPY^•+^, NDI^•−^ and BTA^•^) interactions in solution have been systematically investigated by means of qualitative and quantitative analysis. According to the GKS-EDA results, the nature of RU···CBPQT^n+^ can be summarized as follows:With the variation in charges and multiplicities, in the CRU···CBPQT^n+^ complexes, there are donor–acceptor interactions or radical–closed-shell interactions; in the ORU···CBPQT^n+^ complexes, there are radical pairing or radical–closed-shell interactions. The most important interaction term depends on the charge states in CBPQT^n+^ and RU. The electrostatic term is important when the RU has a negative charge, while desolvation terms always tend to overcome the repulsive electrostatic interactions between the CBPQT cation and RU cation.The nature of radical pairing interactions is different from that of donor–acceptor interactions, but analogous to radical–closed-shell interactions. In radical pairing interactions/radical–closed-shell interactions, the correlation and dispersion terms are always important, while for donor–acceptor interactions, the polarization term is quite large because there is electron transfer between the host and guest.The influence of counterions on the total interactions is also explored, which shows that they can enhance the electrostatic, correlation and dispersion terms while leaving the polarization term almost unchanged.

It is hoped that the understanding of these variable interactions is useful for the design of new recognition sites and multistable molecular switches. It can be found that radical RU is preferred because with the variation in charge and multiplicity, the radical pairing interactions could not result in the electron transfer between the RU and CBPQT, which leads to the large geometrical relaxation. The findings are anticipated to provide a valuable reference for the selection and design of MIMs.

## Figures and Tables

**Figure 1 molecules-28-02057-f001:**
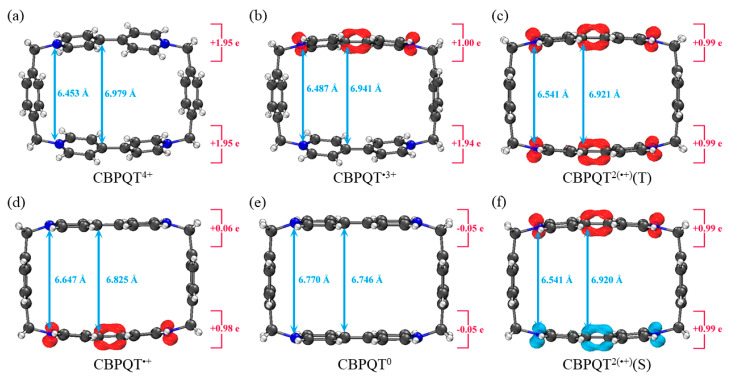
Geometries, NPA charge distribution and spin density distribution of (**a**) CBPQT^4+^, (**b**) CBPQT^•3+^, (**c**) CBPQT^2(•+)^(Triplet), (**d**) CBPQT^•+^, (**e**) CBPQT^0^, (**f**) CBPQT^2(•+)^(Singlet) obtained by ωB97X-D/6-31G* in MeCN solvent. The white, gray and blue spheres represent H, C and N atoms, respectively. The red isosurface represents positive spin density and the blue isosurface is negative.

**Figure 2 molecules-28-02057-f002:**
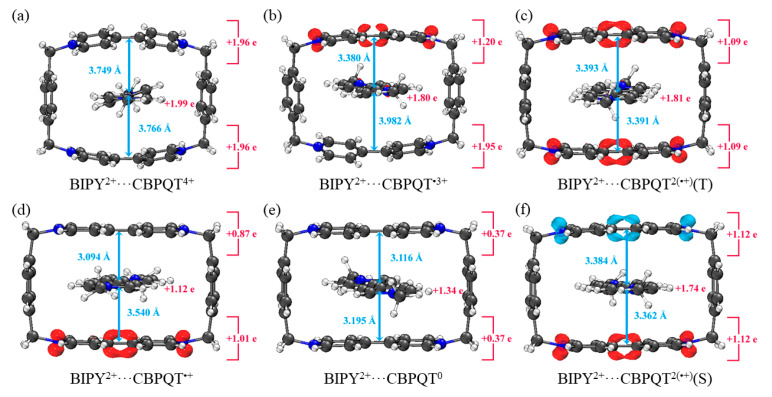
Geometries, NPA charge distribution and spin density distribution of (**a**) BIPY^2+^···CBPQT^4+^, (**b**) BIPY^2+^···CBPQT^•3+^, (**c**) BIPY^2+^···CBPQT^2(•+)^(T), (**d**) BIPY^2+^···CBPQT^•+^, (**e**) BIPY^2+^···CBPQT^0^, (**f**) BIPY^2+^···CBPQT^2(•+)^(S) obtained by ωB97X-D/6-31G* in MeCN solvent.

**Figure 3 molecules-28-02057-f003:**
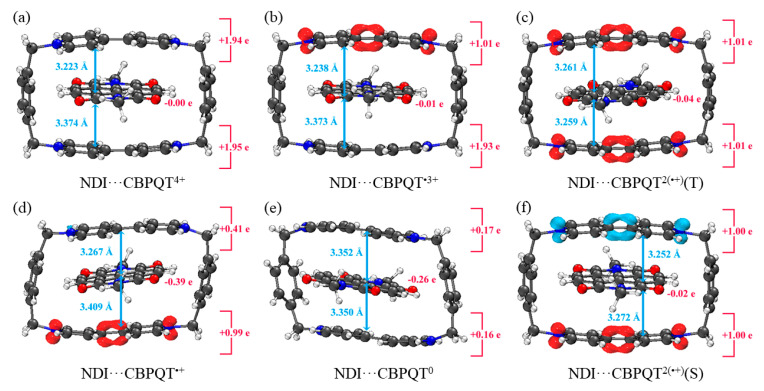
Geometries, NPA charge distribution and spin density distribution of (**a**) NDI···CBPQT^4+^, (**b**) NDI···CBPQT^•3+^, (**c**) NDI···CBPQT^2(•+)^(T), (**d**) NDI···CBPQT^•+^, (**e**) NDI···CBPQT^0^, (**f**) NDI···CBPQT^2(•+)^(S) obtained by ωB97X-D/6-31G* in MeCN solvent.

**Figure 4 molecules-28-02057-f004:**
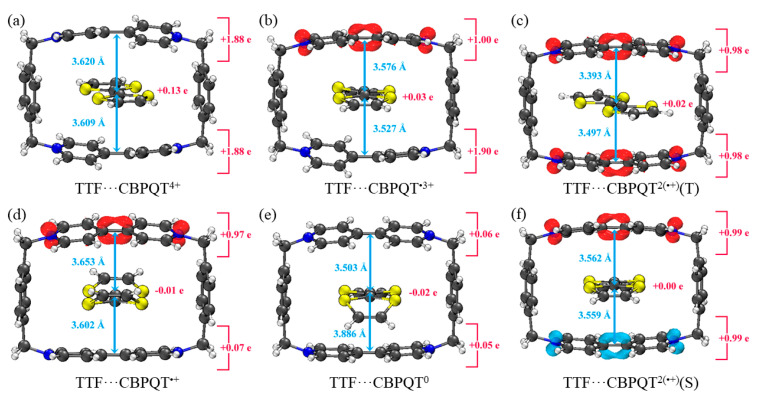
Geometries, NPA charge distribution and spin density distribution of (**a**) TTF···CBPQT^4+^, (**b**) TTF···CBPQT^•3+^, (**c**) TTF···CBPQT^2(•+)^(T), (**d**) TTF···CBPQT^•+^, (**e**) TTF···CBPQT^0^, (**f**) TTF···CBPQT^2(•+)^(S) obtained by ωB97X-D/6-31G* in MeCN solvent.

**Figure 5 molecules-28-02057-f005:**
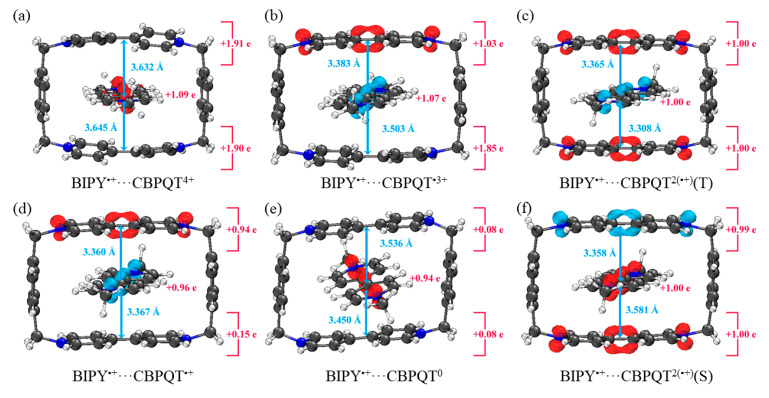
Geometries, NPA charge distribution and spin density distribution of (**a**) BIPY^•+^···CBPQT^4+^, (**b**) BIPY^•+^···CBPQT^•3+^, (**c**) BIPY^•+^···CBPQT^2(•+)^(T), (**d**) BIPY^•+^···CBPQT^•+^, (**e**) BIPY^•+^···CBPQT^0^, (**f**) BIPY^•+^···CBPQT^2(•+)^(S) obtained by ωB97X-D/6-31G* in MeCN solvent.

**Figure 6 molecules-28-02057-f006:**
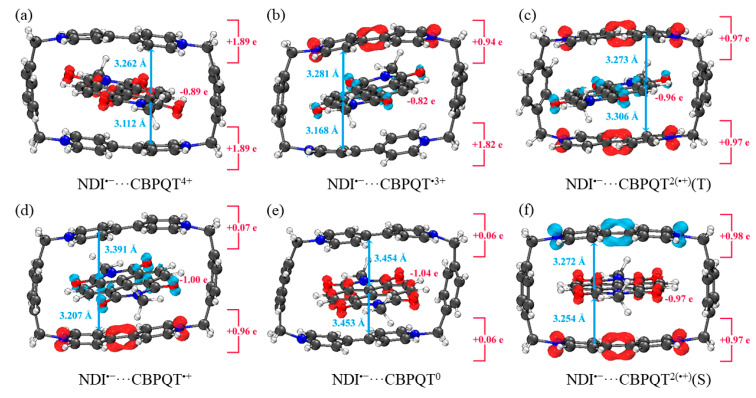
Geometries, NPA charge distribution and spin density distribution of (**a**) NDI^•−^···CBPQT^4+^, (**b**) NDI^•−^···CBPQT^•3+^, (**c**) NDI^•−^···CBPQT^2(•+)^(T), (**d**) NDI^•−^···CBPQT^•+^, (**e**) NDI^•−^···CBPQT^0^, (**f**) NDI^•−^···CBPQT^2(•+)^(S) obtained by ωB97X-D/6-31G* in MeCN solvent.

**Figure 7 molecules-28-02057-f007:**
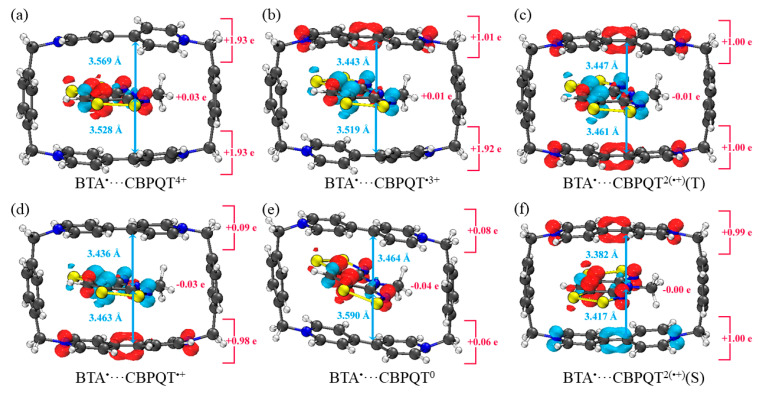
Geometries, NPA charge distribution and spin density distribution of (**a**) BTA^•^···CBPQT^4+^, (**b**) BTA^•^···CBPQT^•3+^, (**c**) BTA^•^···CBPQT^2(•+)^(T), (**d**) BTA^•^···CBPQT^•+^, (**e**) BTA^•^···CBPQT^0^, (**f**) BTA^•^···CBPQT^2(•+)^(S) obtained by ωB97X-D/6-31G* in CPCM solvent.

**Figure 8 molecules-28-02057-f008:**
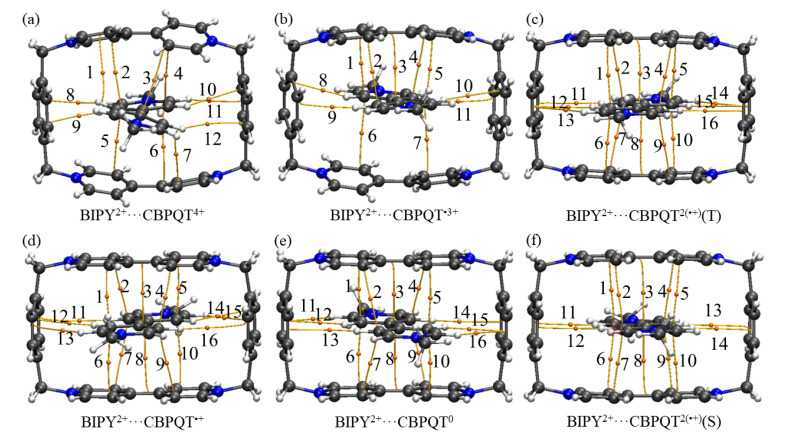
Intermolecular BCPs found in the (**a**) BIPY^2+^···CBPQT^4+^, (**b**) BIPY^2+^···CBPQT^•3+^, (**c**) BIPY^2+^···CBPQT^2(•+)^(T), (**d**) BIPY^2+^···CBPQT^•+^, (**e**) BIPY^2+^···CBPQT^0^, (**f**) BIPY^2+^···CBPQT^2(•+)^(S) complexes.

**Figure 9 molecules-28-02057-f009:**
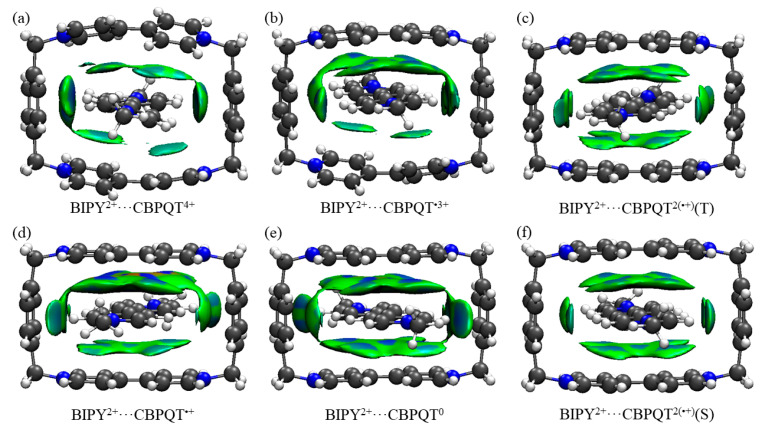
(**a**–**f**) IGM plot iso-surfaces of BIPY^2+^···CBPQT^n+^ (*n* = 0–4) complexes. δ*g*^inter^ = 0.005 a.u. All iso-surfaces are colored according to a BGR scheme over the range −0.03 < sign(*λ*_2_)*ρ* < +0.05 a.u.

**Figure 10 molecules-28-02057-f010:**
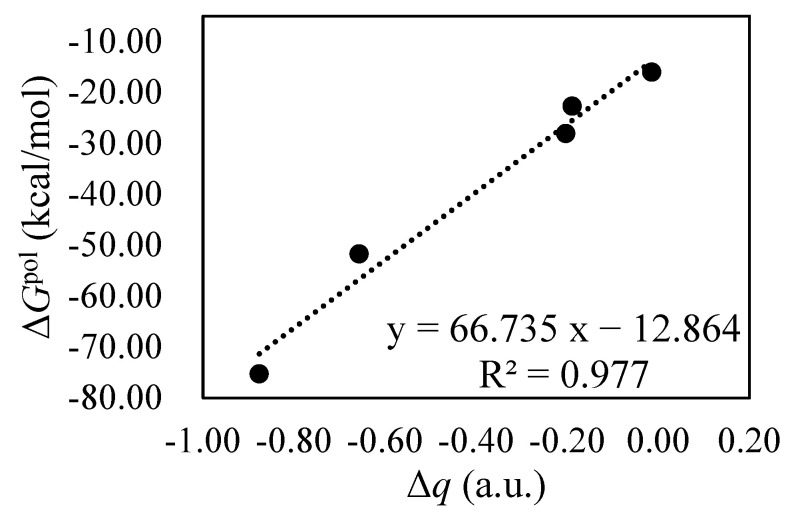
Correlation between the variation in charge (∆*q*) on BIPY^2+^ and polarization interactions of the BIPY^2+^···CBPQT^n+^ complexes in the solution.

**Figure 11 molecules-28-02057-f011:**
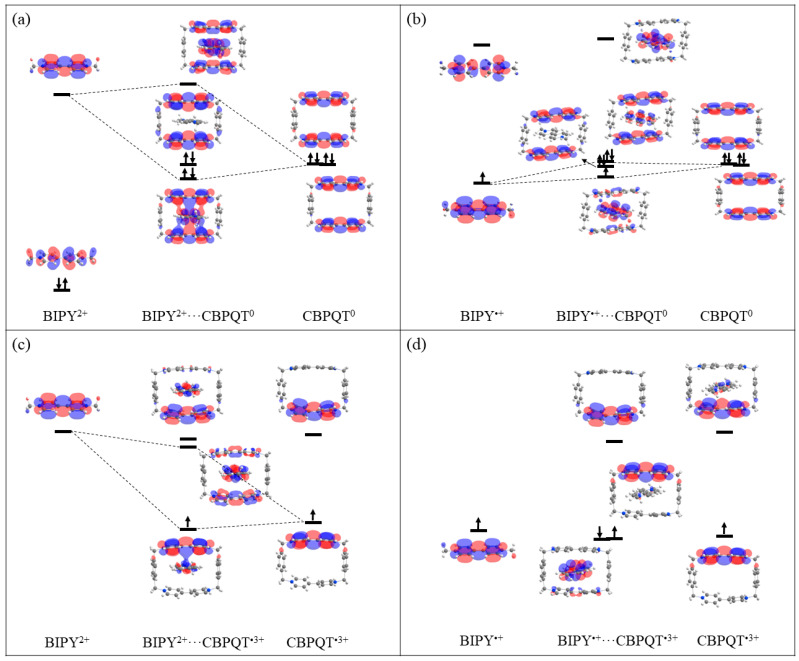
MO diagrams of (**a**) BIPY^2+^···CBPQT^0^, (**b**) BIPY^•+^···CBPQT^0^, (**c**) BIPY^2+^···CBPQT^•3+^, (**d**) BIPY^•+^···CBPQT^•3+^ complexes.

**Figure 12 molecules-28-02057-f012:**
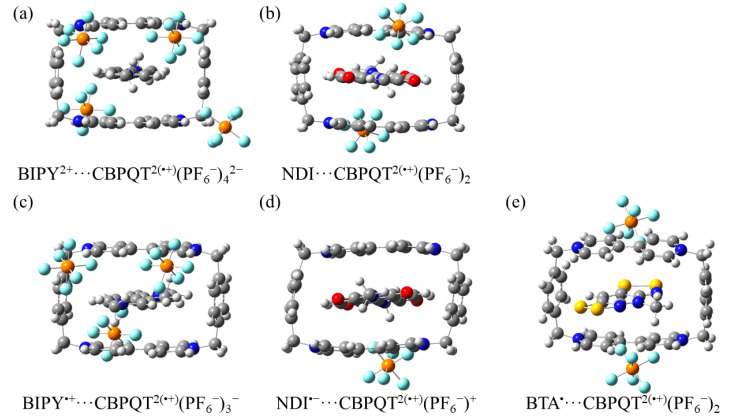
Geometries of (**a**) BIPY^2+^···CBPQT^2(•+)^(PF_6_^−^)_4_^2−^, (**b**) NDI···CBPQT^2(•+)^(PF_6_^−^)_2_, (**c**) BIPY^•+^···CBPQT^2(•+)^(PF_6_^−^)_3_^−^, (**d**) NDI^•−^···CBPQT^2(•+)^(PF_6_^−^)^+^, and (**e**) BTA^•^···CBPQT^2(•+)^(PF_6_^−^)_2_ obtained by ωB97X-D/6-31G* in CPCM MeCN solvent.

**Table 1 molecules-28-02057-t001:** The GKS-EDA results of BIPY^2+^···CBPQT^n+^ complexes obtained by ωB97X-D/6-31+G* in MeCN solvent (kcal/mol).

Complexes	Δ*G*^ele^	Δ*G*^exrep^	Δ*G*^pol^	Δ*G*^desol^	Δ*G*^disp/corr^	Δ*G*^TOT^
BIPY^2+^···CBPQT^4+^	405.32	19.49	−16.03	−373.66	−34.27	0.85
BIPY^2+^···CBPQT^•3+^	296.62	30.41	−28.09	−270.29	−48.01	−19.37
BIPY^2+^···CBPQT^2(•+)^(T)	190.54	34.06	−22.70	−182.87	−50.13	−31.11
BIPY^2+^···CBPQT^2(•+)^(S)	189.98	36.97	−23.22	−181.52	−53.56	−31.34
BIPY^2+^···CBPQT^•+^	70.56	57.56	−75.26	−69.95	−52.53	−69.61
BIPY^2+^···CBPQT^0^	−40.63	62.87	−51.70	22.78	−66.08	−72.77

**Table 2 molecules-28-02057-t002:** The GKS-EDA results of NDI···CBPQT^n+^ complexes obtained by ωB97X-D/6-31+G* in MeCN solvent (kcal/mol).

Complexes	Δ*G*^ele^	Δ*G*^exrep^	Δ*G*^pol^	Δ*G*^desol^	Δ*G*^disp/corr^	Δ*G*^TOT^
NDI···CBPQT^4+^	−22.99	79.11	−15.30	2.71	−69.72	−26.19
NDI···CBPQT^•3+^	−27.04	81.93	−14.63	2.48	−70.44	−27.70
NDI···CBPQT^2(•+)^(T)	−35.53	101.94	−14.21	0.82	−76.87	−23.85
NDI···CBPQT^2(•+)^(S)	−28.93	82.65	−11.77	−0.01	−70.26	−28.33
NDI···CBPQT^•+^	−35.93	95.28	−20.63	0.29	−76.97	−37.97
NDI···CBPQT^0^	−37.99	95.14	−17.22	0.27	−77.22	−37.02

**Table 3 molecules-28-02057-t003:** The GKS-EDA results of TTF···CBPQT^n+^ complexes obtained by ωB97X-D/6-31+G* in MeCN solvent (kcal/mol).

Complexes	Δ*G*^ele^	Δ*G*^exrep^	Δ*G*^pol^	Δ*G*^desol^	Δ*G*^disp/corr^	Δ*G*^TOT^
TTF···CBPQT4+	−34.31	50.09	−8.72	6.11	−44.84	−31.67
TTF···CBPQT•3+	−24.43	38.48	−7.69	5.42	−40.54	−28.77
TTF···CBPQT2(•+)(T)	−26.45	56.43	−7.26	2.83	−48.45	−22.90
TTF···CBPQT2(•+)(S)	−19.11	36.29	−4.86	2.98	−39.44	−24.14
TTF···CBPQT•+	−14.43	37.88	−7.12	1.35	−40.39	−22.71
TTF···CBPQT0	−10.92	35.15	−6.26	0.45	−38.11	−19.69

**Table 4 molecules-28-02057-t004:** The GKS-EDA results of BIPY^•+^···CBPQT^n+^ complexes obtained by ωB97X-D/6-31+G* in MeCN solvent (kcal/mol).

Complexes	Δ*G*^ele^	Δ*G*^exrep^	Δ*G*^pol^	Δ*G*^desol^	Δ*G*^disp/corr^	Δ*G*^TOT^
BIPY^•+^···CBPQT^4+^	187.20	31.77	−8.09	−186.23	−43.82	−19.17
BIPY^•+^···CBPQT^•3+^	135.20	37.84	−11.37	−138.20	−51.08	−27.61
BIPY^•+^···CBPQT^2(•+)^(T)	83.16	42.48	−11.37	−93.51	−54.53	−33.77
BIPY^•+^···CBPQT^2(•+)^(S)	83.86	41.82	−10.13	−93.22	−51.00	−28.67
BIPY^•+^···CBPQT^•+^	31.16	47.04	−14.18	−42.96	−56.91	−35.84
BIPY^•+^···CBPQT^0^	−19.77	44.44	−13.18	3.43	−51.35	−36.42

**Table 5 molecules-28-02057-t005:** The GKS-EDA results of NDI^•−^···CBPQT^n+^ complexes obtained by ωB97X-D/6-31+G* in MeCN solvent (kcal/mol).

Complexes	Δ*G*^ele^	Δ*G*^exrep^	Δ*G*^pol^	Δ*G*^desol^	Δ*G*^disp/corr^	Δ*G*^TOT^
NDI^•−^···CBPQT^4+^	−263.17	117.52	−23.10	207.67	−83.67	−44.75
NDI^•−^···CBPQT^•3+^	−208.11	123.56	−12.20	147.48	−86.41	−35.67
NDI^•−^···CBPQT^2(•+)^(T)	−136.96	92.25	−14.36	99.68	−75.67	−35.06
NDI^•−^···CBPQT^2(•+)^(S)	−136.56	90.66	−13.77	99.75	−73.19	−33.10
NDI^•−^···CBPQT^•+^	−92.29	111.79	−16.96	53.08	−80.92	−25.30
NDI^•−^···CBPQT^0^	−26.00	86.89	−12.03	0.03	−70.36	−21.46

**Table 6 molecules-28-02057-t006:** The GKS-EDA results of BTA^•^···CBPQT^n+^ complexes obtained by ωB97X-D/6-31+G* in MeCN solvent (kcal/mol).

Complexes	Δ*G*^ele^	Δ*G*^exrep^	Δ*G*^pol^	Δ*G*^desol^	Δ*G*^disp/corr^	Δ*G*^TOT^
BTA^•^···CBPQT^4+^	−21.78	55.56	−10.11	4.54	−57.84	−29.64
BTA^•^···CBPQT^•3+^	−23.00	58.80	−10.75	3.83	−58.75	−29.87
BTA^•^···CBPQT^2(•+)^(T)	−23.40	60.15	−8.56	1.07	−59.21	−29.94
BTA^•^···CBPQT^2(•+)^(S)	−24.10	61.56	−8.51	1.34	−59.53	−29.23
BTA^•^···CBPQT^•+^	−25.95	64.72	−11.46	1.49	−59.86	−31.07
BTA^•^···CBPQT^0^	−25.03	61.80	−11.14	−0.38	−56.88	−31.63

**Table 7 molecules-28-02057-t007:** Host–guest interactions after the addition of counterions in BIPY^2+^···CBPQT^2(•+)^, NDI···CBPQT^2(•+)^, BIPY^•+^···CBPQT^2(•+)^,NDI^•−^···CBPQT^2(•+)^, and BTA^•^···CBPQT^2(•+)^ complexes obtained by ωB97X-D/6-31+G* in MeCN solvent (kcal/mol).

	Δ*G*^ele^	Δ*G*^exrep^	Δ*G*^pol^	Δ*G*^desol^	Δ*G*^disp/corr^	Δ*G*^TOT^
BIPY^2+^···CBPQT^2(•+)^(PF_6_^−^)_4_^2−^	−224.84	63.06	−28.08	179.19	−67.61	−78.28
NDI···CBPQT^2(•+)^(PF_6_^−^)_2_	−37.99	94.10	−14.16	2.33	−80.15	−35.87
BIPY^•+^···CBPQT^2(•+)^(PF_6_^−^)_3_^−^	−86.48	65.25	−16.72	53.83	−70.55	−54.67
NDI^•−^···CBPQT^2(•+)^(PF_6_^−^)^+^	−86.84	95.94	−17.82	53.55	−80.37	−35.54
BTA^•^···CBPQT^2(•+)^(PF_6_^−^)_2_	−20.39	54.86	−7.74	0.89	−59.19	−31.59

## Data Availability

The data that supports the findings of this study are available within the article and its Appendix A.

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
