# Peer review of "Radical Pairing Interactions and Donor–Acceptor Interactions in Cyclobis(paraquat-p-phenylene) Inclusion Complexes"

_molecules, 2023, doi:10.3390/molecules28052057_

Round 1

Reviewer 1 Report

This work describes estimation of different types of intermolecular noncovalent, radical-pairing and donor-acceptor interactions in a series of mechanically interlocked molecules. The results represented in the main body of this manuscript give new insights into this field of chemistry, but, as it it was noted by the authors in the conclusions, implicit solvation model does not work good enough for the careful estimation of such energetically low interactions (see for example DOI: 10.1039/D2QO01648F, where the authors shown importance of utilization of explicit solvation model for correct results). Although these systems are very large, I recommend to compare the results utilizing explicit solvation model at least for one model example taken from this work.

In addition, the choice of exact recognition units for this study should be more clearly explained. Why these species were considered among a great series of other units?

Author Response

Reply:

I agree with the referee’s comments. For the CBPQT systems, to consider explicit solvent effect, the most important issue is to consider the effect of counterions. In the revised manuscript, the discussion of interaction between RU and CBPQT2(•+) (PF6-)nm has been added.  

As for the choice of exact recognition units, The second paragraph in page 2 has been modified to explain the selection: “The selection of RU should take the diversity of multiplicity and charge into account. In this work, RUs include bipyridinium radical cation (BIPY•+), bipyridinium dication (BIPY2+), naphthalene-1,8:4,5-bis(dicarboximide) NDI and its radical anion NDI, bis-dithiazolyl radical (BTA) and tetrathiafulvalene (TTF). Thus, these RUs can be grouped as two categories, close-shell RUs (BIPY2+, NDI and TTF) and open-shell RUs (BIPY•+, NDI and BTA)”.  

Reviewer 2 Report

This manuscript is very interesting and well made, and studies the intermolecular interactions between some CBPQTn+ and models of intercalators. The nature of these interactions could be strongly improved by the analysis of bond critical point (BCP) parameters.  

Author Response

Reply:

Thanks! The analysis for bond critical point (BCP) parameters are added in the revised manuscript, the value of BCP parameters are added in supporting materials.

Reviewer 3 Report

The manuscript by Su and co-authors deals with the computational study of noncovalent interactions in inclusion complexes with cyclobis(paraquat-p-phenylene). The standard AIM, IGM and GKS-EDA methods were used to characterize these interactions. This work is technically well done, methods used are adequate. However, this is a quite routine study. Interaction energies were calculated and decomposed, electron density properties were estimated and molecular orbital interactions were briefly analysed. That’s it. Discussion of the trends is not clear. Conclusions on page 13 are obvious and not informative. There are no specific valuable predictions or important interpretations of the interesting experimental issues in conclusions. It is indicated that “ the understanding of these variable interactions is useful to design new recognition sites and multistable molecular switches”, but I do not see how results of this work can help with such a design. This work may be interesting for the specialists working with these compounds, but it is not sufficiently general for Molecules. I suggest the resubmission of the revised version to a more specialized journal on physical chemistry.

Specific comments.

1. Page 1. Definition of a mechanical bond should be provided.

2. Page 3, line 99. expected values -> expectation values

3. It is not clear what G means in this work. Is it Gibbs free energy with the entropic contribution included or this is co-called free energy of solvation which is unfortunately abbreviated as DeltaG in Gaussian and is not the Gibbs free energy?

4. Tables. Is DeltaGdisp/corr a sum of the dispersion and correlation terms? Why not to separate them?

5. Atomic coordinates of the equilibrium geometries should be given in Supplementary Material.

Author Response

The manuscript by Su and co-authors deals with the computational study of noncovalent interactions in inclusion complexes with cyclobis(paraquat-p-phenylene). The standard AIM, IGM and GKS-EDA methods were used to characterize these interactions. This work is technically well done, methods used are adequate. However, this is a quite routine study. Interaction energies were calculated and decomposed, electron density properties were estimated and molecular orbital interactions were briefly analysed. That’s it. Discussion of the trends is not clear. Conclusions on page 13 are obvious and not informative. There are no specific valuable predictions or important interpretations of the interesting experimental issues in conclusions. It is indicated that “ the understanding of these variable interactions is useful to design new recognition sites and multistable molecular switches”, but I do not see how results of this work can help with such a design. This work may be interesting for the specialists working with these compounds, but it is not sufficiently general for Molecules. I suggest the resubmission of the revised version to a more specialized journal on physical chemistry.

Reply:

Interactions between RU and CBPQT are complex with the variation of charges and multiplicities in RU and CBPQT. In the revised manuscript, we have added the discussion of counterions’ effect, re-written the methodology section and conclusion section, to emphasize the scientific significance. In the last paragraph of conclusion, our advice for the MIM design is added.

Specific comments.

  1. Page 1. Definition of a mechanical bond should be provided.

Reply: Thanks. The definition has been added.

  1. Page 3, line 99. expected values -> expectation values

Reply:

Thanks. It is corrected.

  1. It is not clear what G means in this work. Is it Gibbs free energy with the entropic contribution included or this is co-called free energy of solvation which is unfortunately abbreviated as DeltaG in Gaussian and is not the Gibbs free energy?

Reply:

The methodology of GKS-EDA has been re-written to explain the meaning of “G”.

  1. Tables. Is DeltaGdisp/corr a sum of the dispersion and correlation terms? Why not to separate them?

Reply

Yes. The DeltaGdisp/corr a sum of the dispersion and correlation terms. The definition has been added in the methodology section.

  1. Atomic coordinates of the equilibrium geometries should be given in Supplementary Material.

Reply

The coordinates have been added in Supplementary Material.

Round 2

Reviewer 3 Report

The manuscript was a bit improved but I still think that this work is more suitable for a more specialized journal on physical chemistry. Meanwhile, I leave the decision on the judgement of the Editor.